# A Simple Dynamic Learning Rate Tuning Algorithm For Automated Training of DNNs

## Abstract

Training neural networks on image datasets generally require extensive experimentation to find the optimal learning rate regime. Especially, for the cases of adversarial training or for training a newly synthesized model, one would not know the best learning rate regime beforehand. We propose an automated algorithm for determining the learning rate trajectory, that works across datasets and models for both natural and adversarial training, without requiring any dataset/model specific tuning. It is a stand-alone, parameterless, adaptive approach with no computational overhead. We theoretically discuss the algorithm's convergence behavior. We empirically validate our algorithm extensively. Our results show that our proposed approach *consistently* achieves top-level accuracy compared to SOTA baselines in the literature in natural as well as adversarial training.

## 1 Introduction

Deep architectures are generally trained by minimizing a non-convex loss function via underlying optimization algorithm such as stochastic gradient descent or its variants. It takes a fairly large amount of time to find the best suited optimization algorithm and its optimal hyperparameters (such as learning rate, batch size etc.) for training a model to the desired accuracy, this being a major challenge for academicians and industry practitioners alike. Usually, such tuning is done by initial configuration optimization through grid search or random search (Bergstra et al., 2011; Snoek et al., 2012; Thornton et al.) . Recent works have also formulated it as a bandit problem (Li et al., 2017).

However, it has been widely demonstrated that hyperparameters, especially the learning rate often needs to be dynamically adjusted as the training progresses, irrespective of the initial choice of configuration. If not adjusted dynamically, the training might get stuck in a bad minima, and no amount of training time can recover it. In this work, we focus on learning rate which is the foremost hyperparameter that one seeks to tune when training a deep learning model to get favourable results.

Certain auto-tuning and adaptive variants of SGD, such as AdaGrad (Duchi et al., 2011), Adadelta (Zeiler, 2012), RMSProp (Tieleman & Hinton, 2012), Adam (Kingma & Ba, 2015) among others have been proposed that automatically adjust the learning rate as the training progresses, using functions of gradient. Yet others have proposed fixed learning rate and/or batch size change regimes (Goyal et al., 2017), (Smith et al., 2018) for certain data set and model combination.

In addition to traditional natural learning tasks where a good LR regime might already be known from past experiments, adversarial training for generating robust models is gaining a lot of popularity off late. In these cases, tuning the LR would generally require time consuming multiple experiments, since the LR regime is unlikely to be known for every attack for every model and dataset of interest[1]. Moreover, new models are surfacing every day courtesy the state-of-the-art model synthesis systems, and new datasets are also becoming available quite often in different domains such as healthcare, automobile industy etc. In each of these cases, no prior LR regime would be known, and would require considerable manual tuning in the absence of a universal method, with demonstrated effectiveness over a wide range of tasks, models and datasets.

Wilson et al. (2017) observed that solutions found by existing adaptive methods often generalize worse than those found by non-adaptive methods. Even though initially adaptive methods might

---

[1]For example, one can see a piecewise LR schedule given by Madry et al. (2018) at `https://github.com/MadryLab/cifar10_challenge/blob/master/config.json` for a particular model.

display faster initial progress on the training set, their performance quickly plateaus on the test set, and learning rate tuning is required to improve the generalization performance of these methods. For the case of SGD with Momentum, learning rate (LR) step decay is very popular (Goyal et al., 2017),(Huang et al., 2017), ReduceLRonPlateau[2]. However, in certain junctures of training, increasing the LR can potentially lead to a quick, further exploration of the loss landscape and help the training to escape a sharp minima (having poor generalisation Keskar et al. (2016)). Further, recent works have shown that the distance traveled by the model in the parameter space determines how far the training is from convergence Hoffer et al. (2017). This inspires the idea that increasing the LR to take bigger steps in the loss landscape, while maintaining numerical stability might help in better generalization.

The idea of increasing and decreasing the LR periodically during training has been demonstrated by Smith (2017); Smith & Topin (2017) in their cyclical learning rate method (CLR). This has also been shown by Loshchilov & Hutter (2016), in Stochastic Gradient Descent with Warm Restarts (SGDR, popularly referred to as Cosine Annealing with Warm Restarts). In CLR, the LR is varied periodically in a linear manner, between a maximum and a minimum value, and it is shown empirically that such increase of learning rate is overall beneficial to the training compared to fixed schedules. In SGDR, the training periodically restarts from an initial learning rate, and then decreases to a minimum learning rate through a cosine schedule of LR decay. The period typically increases in powers of 2. The authors suggest optimizing the initial LR and minimum LR for good performance.

Schaul et al. (2013) had suggested an adaptive learning rate schedule that allows the learning rate to increase when the signal is non-stationary and the underline distribution changes. This is a computationally heavy method, requiring computing the Hessian in an online manner.

Recently, there has been some work that explore gradients in different forms for hyperaparameter optimization. Maclaurin et al. (2015) suggest an approach by which they exactly reverse SGD with momentum to compute gradients with respect to all continuous learning parameters (referred to as hypergradients); this is then propagated through an inner optimization. Baydin et al. (2018) suggest a dynamic LR-tuning approach, namely, hypergradient descent, that apply gradient-based updates to the learning rate at each iteration in an online fashion.

We propose a new algorithm to automatically determine the learning rate for a deep learning job in an autonomous manner that simply compares the current training loss with the best observed thus far to adapt the LR. The proposed algorithm works across multiple datasets and models for different tasks such as natural as well as adversarial training. It is an 'optimistic' method, in the sense that it increases the LR to as high as possible by examining the training loss repeatedly. We show through rigorous experimentation that in spite of its simplicity, the proposed algorithm performs surprisingly well as compared to the state-of-the-art.

**Our contributions:**

- We propose a novel, and simple algorithmic approach for autonomous, adaptive learning rate determination *that does not require any manual tuning*, inspection, or pre-experimental discovery of the algorithmic parameters.

- Our proposed algorithm works across data sets and models *with no customization* and reaches higher or comparable accuracy as standard baselines in literature in the same number of epochs on each of these datasets and models. It *consistently* performs well, finding stable minima with good generalization and converges *smoothly*.

- Our algorithm works very well for adversarial learning scenario along with natural training as demonstrated across different models and datasets.

- We provide extensive empirical validation of our algorithm and convergence discussion.

## 2 PROPOSED METHOD

We propose an autonomous, adaptive LR tuning algorithm 1 towards determining the LR trajectory during the course of training. It operates in two phases: Phase 1: Initial LR exploration, that strives

---

[2]For eg. `https://www.tensorflow.org/api_docs/python/tf/keras/callbacks/ReduceLROnPlateau`.

to find a good starting LR; Phase 2: Optimistic Binary Exploration. The pseudocode is provided at Algorithm 1. For the rest of the paper, we refer to the Automated Adaptive Learning Rate tuning algorithm as AALR in short.

---

**Algorithm 1** Automated Adaptive Learning Rate Tuning Algorithm (AALR) for Training DNNs

---

**Require:** Model $\theta$, $N$ Training Samples $(x_i, y_i)_{i=1}^N$, Optimizer SGD, Momentum=0.9, Weight Decay, Batch size, Number of epochs $T$, Loss Function $J(\theta)$. Initial LR $\eta_0 = 0.1$.
**Ensure:** Learning Rate $\eta_t$ at every epoch $t$.
 1: Initialize: $\theta$, SGD with LR= $\eta_0$, best loss $L^* \leftarrow J(\theta)$ (forward pass through initial model).
 2: **PHASE 1:** Start Initial LR Exploration.
 3: Set patience $p \leftarrow 10$, patience counter $i \leftarrow 0$, epoch number $t \leftarrow 0$
 4: **while** $i < p$ **do**
 5:     Evaluate new loss $L \leftarrow J(\theta)$ after training for an epoch. Increment $i$ and $t$ by 1 each.
 6:     **if** $L > L*$ or $L$ is NAN **then**
 7:         Halve LR: $\eta_0 \leftarrow \eta_0/2$.
 8:         Reload $\theta$, reset optimizer with LR= $\eta_0$, and reset counter $i = 0$.
 9:     **else**
10:         $L^* \leftarrow L$ (Update best loss).
11:     **end if**
12: **end while**
13: Save checkpoint $\theta$ and $L*$.
14: **PHASE 2:** Start Optimistic Binary Exploration
15: Double LR $\eta_t \leftarrow 2\eta_0$. Patience $p \leftarrow 1$.
16: **while** $t < T$ **do**
17:     Train for $p$ epochs. Increment epoch number $t$.
18:     Evaluate new loss $L \leftarrow J(\theta)$.
19:     **if** $L$ is NAN **then**
20:         Halve LR: $\eta_{t+1} \leftarrow \eta_t/2$.
21:         Load checkpoint $\theta$ and $L*$. Reset optimizer with LR= $\eta_{t+1}$.
22:         Double patience $p_{t+1} \leftarrow 2p_t$. Continue.
23:     **end if**
24:     **if** $L < L^*$ **then**
25:         Update $L^* \leftarrow L$. Save checkpoint $\theta$ and $L*$.
26:         Double LR $\eta_{t+1} \leftarrow 2\eta_t$. Set patience $p = 1$.
27:     **else**
28:         Train for another $p$ epochs. Increment epoch number $t$.
29:         Evaluate new loss $L \leftarrow J(\theta)$.
30:         **if** $L < L^*$ **then**
31:             Update $L^* \leftarrow L$. Save checkpoint $\theta$ and $L*$.
32:             Double LR $\eta_{t+1} \leftarrow 2\eta_t$. Set patience $p = 1$.
33:         **else**
34:             Halve LR: $\eta_{t+1} \leftarrow \eta_t/2$.
35:             Double patience $p_{t+1} \leftarrow 2p_t$.
36:             **if** $L$ is NAN **then**
37:                 Load checkpoint $\theta$ and $L*$. Reset optimizer with LR= $\eta_{t+1}$.
38:             **end if**
39:         **end if**
40:     **end if**
41: **end while**

---

The notation used in the following description is as follows. Patience: $p$, Learning rate: $\eta$, best loss $L^*$, current loss $L$. Model $\theta$, Loss function $J(\theta)$. $L^*$ is initialized as the $J(\theta)$ after initializing the model, before training starts.

**Phase 1: Initial LR exploration**
Phase 1 starts from an initial learning rate $\eta = 0.1$, and patience $p = 10$. It trains for an epoch, evaluates the loss $L$, and compares to the best loss $L^*$. If $L < L^*$, the $L^*$ is updated, and it continues training for another epoch. Otherwise, the model $\theta$ is reloaded and re-initialized, LR

is halved $\eta = \eta/2$, and optimizer is reset with the new LR. The patience counter is reset. This continues till a stable LR is determined by the algorithm, in which it trains at this LR for $p$ epochs. The loss $L^*$, the model $\theta$ and the optimizer state after Phase 1 is saved in a checkpoint.

**Phase 2: Optimistic Binary Exploration**
In this phase, AALR keeps the learning rate $\eta$ as high as possible for as long as possible at any given state of the training. Phase 2 starts by doubling LR to $2\eta$, and setting $p = 1$. After training for $p+1$ epochs, firstly AALR checks if the loss is NAN. In this case, the checkpoint (model $\theta$ and optimizer) corresponding to the best loss along with the best loss value $L^*$ are reloaded. Then LR is halved $\eta = \eta/2$, patience is doubled, and the training continues. If instead, the loss is observed to decrease compared to the best loss, $L < L^*$, then $L^*$ is updated, and the corresponding model $\theta$, optimizer and $L^*$ are updated in checkpoint. This is followed by doubling the LR $\eta = 2\eta$, resetting $p$ to 1 and continuing training for the next $p+1$ epochs.

On the other hand, if $L \geq L^*$, AALR trains for another $p+1$ epochs and check the loss $L$. This is because as informally stated before, AALR is 'optimistic' and persists in the high LR for some more time. (In case, the newly evaluated loss is NAN, the previous approach is followed.) However, if the new loss $L \geq L^*$, then AALR *finally* lowers the LR. AALR halves the LR $\eta = \eta/2$, doubles patience $p = 2p$, and continues training for $p+1$ epochs. If however, the loss had decreased, $L < L^*$, the previous approach is followed: i.e., it doubles the LR $\eta = 2\eta$, resets the patience $p = 1$, updates best loss and checkpoint, and repeats training for $p+1$ epochs. The above cycle repeats till the stopping criterion is met. For ease of exposition, the pseudocode is given in Algorithm 1

## 3 MOTIVATION AND RELATED WORK

Increasing the LR optimistically can potentially help the training to escape saddle points that slow down the training, as well as find flatter minima with good generalization performance. This is inspired mainly from the following observations in the literature.

Dauphin et al. (2014) suggest that saddle points slow down the training of deep networks. Xing et al. (2018) states that SGD moves in valley like regions of the loss surface in deep networks by jumping from one valley wall to another at a height above the valley floor which is determined by the LR. Large LR can help in generalization by helping SGD to quickly cross over the valley floor as well as its barriers, to travel far away from the initialization point in a short time. Similarly, Hoffer et al. (2017) describe the initial training phase as a high-dimensional "random walk on a random potential" process, with an "ultra-slow" logarithmic increase in the distance of the weights from their initialization.

From the above discussion, it seems that if one could increase the step size or LR continuously (as long as stability is maintained), it might considerably speed up the increase in distance of the weights from the initialization point, making the initial ultra-slow diffusion process faster. In this way, further exploration of the loss landscape might be possible, leading to better generalization.

The idea of increasing the LR has been explored by algorithms like SGDR and CLR. In SGDR, the LR is reset to a high value in a periodic manner; this is referred to as warm restart. After this, the LR decays to a low value following a cosine annealing schedule. In CLR, the LR is increased and decreased linearly in a periodic manner. While the regular increase in LR in most cases, probably helps generalization and helps in finding flatter minima, they follow a preset method, that does not depend on the training state or progress. Detection of convergence also becomes difficult due to heavy fluctuation in the training output (which happens due to the periodic nature of these methods). Moreover, the authors of each of these methods suggest tuning the parameters of the algorithm for better performance. Even though Schaul et al. (2013) suggest a method that uses information about the state and distribution, it is computationally heavy method. Similarly, hypergradient descent due to Baydin et al. (2018) requires additional computation of gradients. Moreover, it requires tuning of initial LR and introduces additional hyperparameters to be tuned, such as the learning rate for the LR itself.

We propose the simple idea of exploring LR in a binary fashion, without requiring any parameter tuning. This is an adaptive LR tuning algorithm that tries to *follow the training state* and set the LR accordingly. Increasing LR for better generalization through exploration (and also, potential acceleration of initial phase of SGD) are the main motivations for the optimistic doubling. At the

same time, once SGD is in the vicinity of a good minimum, LR might need to be reduced to access the valley. Hence, if the algorithm observes that the loss is not reducing even after a few 'patience' iterations, it halves the LR. The reduction is kept conservative at $1/2$ to encourage finding flatter minima.

The automated adaptive LR algorithm we propose achieves good generalization in all cases, including adversarial scenario, and converges smoothly in roughly the same time as LR-tuned SGD baselines available in the literature and community.

## 4 CONVERGENCE DISCUSSION

Convergence analysis of SGD typically requires that the sequence of step sizes, or, learning rates used during training satisfy the following conditions: $\sum_{t=1}^{\infty} \eta_t = \infty$ and $\sum_{t=1}^{\infty} \eta_t^2 < \infty$.

Consider an optimal stochastic gradient approach OPT that any point in time has oracle access to (and applies) the highest value of learning rate, that would be amenable for good training (ensuring fast convergence and good generalization). The sequence of LRs chosen by OPT satisfy the above condition. The sequence ensure that OPT will converge (to a good generalization), at the same time, the convergence is the fastest since the step sizes or LRs are kept as high as possible. Let the LR of OPT at any epoch $t$ be denoted as $\rho_t$.

One can define OPT as the following:

**Definition:** An optimal oracle SGD, with LR $\rho_t$ at any epoch $t$, such that the following properties hold:

1. $\sum_{t=1}^{\infty} \rho_t = \infty$,
2. $\sum_{t=1}^{\infty} \rho_t^2 < \infty$,
3. Any first order stochastic gradient-based algorithm that has the same location in parameter space as OPT at the start of an epoch $t$ must have LR $\leq \rho_t$, otherwise training will diverge here onward (in other words, gradients will explode and the training of the network cannot be recovered from here).
4. Any first order stochastic gradient based algorithm starting from the same location in parameter space as OPT and achieving similar generalization for a given training task, will require at least as many epochs as OPT for convergence.

We will compare AALR with OPT, and show that the sequence of LRs chosen by AALR follow the sequence of LRs of OPT with a bounded delay under some assumptions. We also show that divergence will not happen.

A typical well-tuned SGD algorithm can be thought to be a proxy for OPT for a given scenario, and hence this analysis will bound the convergence time of AALR with respect to LR-tuned SGD for the same problem.

In a typical step decay LR-regime for SGD, the LR does not increase, but generally decreases at certain intervals by some factor $\gamma \in \{2, \ldots, 10\}$. For standard LR schedules, one can see that the following rule-of-thumb holds: the number of epochs $\Delta$ in between two consecutive LR changes is directly proportional to $\gamma$. In fact one can see that for standard regimes, $\Delta \geq c\gamma$, where $c \geq 2$. (For example, change by a factor of 2 happens at every 5 epochs or more, or, change by a factor of 10 happens every 30 epochs or more). Such typical LR regimes are often designed out of observing of loss plateauing. We assume that OPT has a similar behaviour in the following analysis.

### 4.1 BOUNDING THE DELAY IN CONVERGENCE DUE TO DOUBLING

Let the LR of OPT at any epoch $t$ be denoted as $\rho_t$ and that of AALR be denoted as $\eta_t$. We assume that Phase 1 has estimated a stable initial LR $\eta_0 \leq \rho_0$, and that both AALR and OPT are roughly in the same space in the loss valley at the start of Phase 2, denoted as epoch 1 (for simplicity). In the following, we refer to decrease in loss compared to the best observed loss thus far as an improvement in state.

Assuming that the loss surface is smooth, the loss will continue to decrease for AALR, as long as $\eta_t \leq \rho_t$, and it will start increasing otherwise.

We first argue that AALR will not diverge. From Algorithm 1, it can be seen that every time state improves, the checkpoint is updated. If and when, due to doubling (or due to initial LR), loss diverges and goes to NAN, the last checkpoint is reloaded, LR is halved and training continues. This will continue till a stable LR is reached, and loss is no longer NAN. In this way, AALR can avoid losing way due to exploding gradients, caused by undue increase in LR.

Now, let us consider the case, when OPT has increased its LR. AALR, by design is always optimistically doubling the LR whenever state improves. For an increase in $\rho_t$ by a factor $\gamma$, it can be seen that AALR will require $2\log_2 \gamma$ epochs. This is because, when state improves, patience $p$ will be reset to 1, LR will be doubled, and the state will be checked again after training for $p + 1 = 2$ epochs.

We would next show that AALR reaches the same or lower LR as OPT (with some delay) every time OPT reduces LR.

AALR starts with $\eta_1 = 2\rho_0$ and trains for $p + 1 = 2$ epochs and checks the state. If OPT has maintained $\rho(t)$ at $\rho_0$, then state will not improve. AALR will train for another 2 epochs, and then reduce LR by half, and double the patience. It trains at this LR for $2p + 1 = 3$ epochs. Therefore, effective training for AALR is for 3 epochs out of the 7 epochs it spent. Now if the state improves, AALR would double the LR and the above cycle would repeat till we come to the state where OPT needs to reduce the LR for making progress. Let there be $k$ such cycles, such that OPT has trained for $3(k-1) < q \leq 3k$ epochs and AALR has trained for $7k$ epochs to arrive at roughly the same location in parameter space (assuming bounded gradients) and both have the same LR.

Now, let OPT reduce its LR by $\gamma$, i.e., $\rho_{q+1} = \rho_p/\gamma$ (For simplicity, let us assume that $\gamma$ is a power of 2). AALR would be first doubling the LR to $\eta_{7k+1} = 2\eta_{7k} = 2\rho_q$ (since its state was improving till $7k$ epochs), and patience $p$ will be reset to 1. It will need to reduce its LR $1 + \log_2 \gamma$ times before it observes an improves in state (by assumption on OPT maintaining the highest possible LR for training progress). It will train for $2(p+1) = 2(1+1) = 4$ epochs, then halve the LR, double $p$, train for $2(p+1) = 2(2+1) = 6$ epochs, and repeat this for $n = 1 + \log_2 \gamma$ times. One can see by induction that AALR will be spending a total of $N_{AALR} = \sum_{i=0}^{n-1} 2(2^i + 1) = 2n + 2^{n+1} - 1$ epochs. At this time, $p = 2^{n-1}$. Now, AALR will train for $2(p+1) = 2^n + 2$ epochs at this LR, after which it will observe an improvement in state. Note that by the earlier observation regarding typical LR regimes and the behavior of OPT, OPT would train for at least $\sim 2\gamma = 2^n$ epochs at this new LR. Hence, AALR has trained for a total of roughly twice the number of epochs as OPT, and at the new LR for roughly the same number of epochs as OPT. Therefore, both are now at a similar location in parameter and loss space. After this AALR would again double the LR, and the earlier cycle would repeat for another $k'$ times, such that OPT would have trained for $3(k'-1) < p' \leq 3k'$ epochs and AALR would have trained for $7k'$ epochs till the next LR change happens in OPT. Therefore, one can see that AALR would take $\sim 2$ times the number of epochs as OPT to reach the same or lower LR, every time OPT lowers the LR.

Since the LR sequence of AALR follows the LR sequence of OPT with some *finite delay*, it can be argued that the following convergence requirements on the LR sequence hold for AALR: (1) $\sum_{t=1}^{\infty} \eta_t = \infty$, and, (2) $\sum_{t=1}^{\infty} \eta_t^2 < \infty$. In practice, we observe that AALR converges in around the same time as LR-tuned SGD.

## 5 EXPERIMENTS

We trained with AALR on several model-datasets combinations, in multiple scenarios such as natural training, as well as adversarial training. We observed that AALR achieve similar or better accuracy as the state-of-the-art baselines.

We have compared to standard SOTA (SGD or other) LR tuned values reported in the literature and with three other adaptive LR tuning algorithms, SGDR (Cosine Scheduling with Warm Restarts), CLR (Cyclic Learning Rates), ADAM, and Hypergrad (Baydin et al., 2018) (few runs were not complete). Since the principle claim of AALR is that it is a completely autonomous adaptive approach that *does not require any tuning*, for *fair comparison*, we have *not tuned* the parameters of

any other adaptive approaches compared with. Since AALR does not have any *tunable* parameters by design, sensitivity analyses experiments were not performed for AALR.

Additionally, we have done some experiments examining the relationship of batchsize andor entropy with the learning rate. We find the AALR finds LR trajectory that roughly obey the square root rule as suggested by Hoffer et al. (2017) and

## 5.1 SETTINGS, DATASETS AND MODELS

Experiments were done in PyTorch in x86 systems using 6 cores and 1 GPU. Where baselines for SGDR (Loshchilov & Hutter, 2016) and CLR are not available in literature, the PyTorch provided implementations of the corresponding LR schedulers with default settings were used (available here `https://pytorch.org/docs/stable/_modules/torch/optim/lr_scheduler.html`. For Hypergrad (we have tried SGDHD optimizer), we have used the code available at `https://github.com/gbaydin/hypergradient-descent`.

We have tested on datasets CIFAR10 and CIFAR100 using standard data augmentation for both, on models Resnet-18, WideResnet-28-10 with dropout and cutout, WideResnet-34-10 (for adversarial scenario only), SimplenetV1, and Vgg16 with and without batch normalization. For Resnet-18, WideResNets and Vgg16 models, we ran all algorithms for 200 epochs at a batch size of 128 for both datasets. For SimplenetV1, we ran for 540 epochs and used batch size of 100 and 64 for CIFAR10 and CIFAR100 respectively for all algorithms. The code for SimpleNetV1 was obtained from `https://github.com/Coderx7/SimpleNet_Pytorch`. The code for Resnet-18 and WideResnets was obtained from `https://github.com/uoguelph-mlrg/Cutout`. The code for Vgg16 was obtained from `https://github.com/chengyangfu/pytorch-vgg-cifar10/blob/master/main.py`. We use cross entropy loss in all cases.

The batchsize and number of epochs used in each case was **as per the baselines in literature**, i.e., the reported SOTA results for LR-tuned SGD. The baseline (and in where available, adaptive algorithms') results were reported from the cited sources. For the relationship of batchsize with learning rate, we use FMNIST on ResNet-20.

## 5.2 OBSERVATIONS

Our experiments comprehensively show that AALR is a state-of-the-art automated adaptive LR tuning algorithm that works universally across models-datasets for both natural and adversarial training. It is either better or comparable to LR-tuned baselines and other adaptive algorithms *uniformly and consistently*, with a *smooth convergence* behavior. It effective across batchsizes, without requiring any tuning. This makes the case that for new models andor datasets, AALR should be a reliable LR algorithm of choice, in the absence of any prior tuning or experimentation.

For the adaptive algorithms we compared with, SGDR though performs comparable with AALR in most cases of or natural training, it catastrophically failed for at least two natural training cases (which indicates it require tuning of either initial LR or some other parameters, and hence not a completely stand alone automated approach) and moreover, it generally did significantly worse than AALR for adversarial training. CLR achieved slightly lower accuracy compared to AALR in most cases of natural training, and in adversarial training its performance fluctuated on a case by case basis. ADAM generally converged to lower accuracy and significantly lower in adversarial scenario, and furthermore, it catastrophically failed in two cases of natural training, which shows it requires extensive tuning of parameters.

AALR was consistently top-level in every case, which makes the case for its universal and reliable applicability, especially when new models/datasets/training tasks surface for which prior tuning or information is not available.

## 5.3 NATURAL TRAINING

The baseline values reported have the following sources:

- Resnet-18, CIFAR10 as reported by DeVries & Taylor (2017),

- WideResnet-28-10, CIFAR10 and CIFAR100 baseline as reported by Zagoruyko & Komodakis (2016), and corresponding SGDR values as reported by Loshchilov & Hutter (2016)

- WideResnet-28-10 with Dropout and Cutout, CIFAR10 and CIFAR100 as reported by De-Vries & Taylor (2017)

- SimplenetV1, CIFAR10 and CIFAR100 as reported at HasanPour et al. (2016) and `https://github.com/Coderx7/SimpleNet_Pytorch`

- Vgg16 with and without Batch Normalization for CIFAR10 as reported at `https://github.com/chengyangfu/pytorch-vgg-cifar10` and `http://torch.ch/blog/2015/07/30/cifar.html` (the former values are higher). (CIFAR100 for Vgg16 values were not reported at these places, hence not provided in the table.)

| Model | Baseline | AALR | SGDR | CLR | ADAM | Hypergrad |
|---|---|---|---|---|---|---|
| Resnet-18 | 95.28 | 94.94 | 94.81 | 93.85 | 93.07 | 84.48 |
| SimpleNet-V1 | 95.51 | 95.17 | 95.44 | 93.66 | 93.99 | 87.01 |
| Vgg16 | 91.4, 92.63, | 92.16 | 10.00 | 91.95 | 10.00 (78.46*) | 10.00 |
| Vgg16-BN | 92.45, 93.86 | 93.23 | 93.56 | 92.65 | 91.48 | 85.05 |
| WRN-28-10 (Dropout) | 96.00 | 95.75 | 95.91 | 95.34 | 94.01 | - |
| WRN-28-10 (Dropout + Cutout) | 96.92 | 96.44 | 96.6 | 95.42 | 95.46 | - |

Table 1: Natural Training on CIFAR10. Comparison of test accuracy of model trained with AALR with baselines and with those obtained by training using different adaptive learning rate techniques on various models.*:With AALR, ADAM drastically improves. Peak accuracy 78.46 from 10.

| Model | Baseline | AALR | SGDR | CLR | ADAM |
|---|---|---|---|---|---|
| WRN-28-10 (Dropout) | 79.96 | 80.45 | 80.26 | 78.63 | 73.67 |
| SimpleNet-V1 | 78.51 | 78.21 | 77.47 | 74.02 | 73.48 |
| Vgg16 | - | 65.03 | 10.00 | 67.79 | 10.00 |

Table 2: Natural Training on CIFAR100: Comparison of test accuracy of model trained with AALR with baselines and with those obtained by training using different adaptive learning rate techniques on various models.

## 5.4 ADVERSARIAL TRAINING

Here we outline the results and observations from adversarial training. We observe that AALR is particularly effective in Adversarial Training and achieves (to the best of our knowledge) state-of-the-art adversarial test accuracy for FGSM attack in a White Box model[3].

It generally does significantly better compared to the other adaptive algorithms compared with and convergence is easier to detect, unlike the other methods. It would be interesting to explore theoretical justification regarding the effectiveness of AALR in first-order adversarial training, and it might be related to the loss landscape of the min-max saddle point problem defined by Madry et al. (2018).

In the process of these experiments, we discover that SimpleNetV1 is a very effective adversarially strong model with respect to CIFAR10, when trained especially with AALR for FGSM attack ($\epsilon = 8/255$ and $\alpha = 2/255$) in White Box model. This is a light weight model. Adversarial training generally being a compute heavy and time consuming process, becomes much easier and faster with it. The effectiveness of AALR in training models for different and new scenarios is clearly

---

[3]For readers unfamiliar with adversarial training and FGSMPGD attacks, details can be found in Madry et al. (2018). We have given a short description here. Adversarial training aims at solving the min-max saddle point problem defined as: $\min_\theta \rho(\theta)$ where $\rho(\theta) = E_{(x,y)\sim\mathcal{D}}[\max_{\delta\in\mathcal{S}} L(\theta, x + \delta, y)]$ in contrast to standard training which simply aims at minimizing $E_{(x,y)\sim\mathcal{D}}[L(\theta, x, y)]$. FGSM simply perturbs the data as $x + \varepsilon\ sgn(\nabla_x L(\theta, x, y))$, whereas PGD does it for $k$ steps iteratively as follows: $\Pi_{x+\mathcal{S}}(x^t + \alpha\varepsilon sgn(\nabla_x L(\theta, x, y)))$. White Box attack refers to the attacker having access to the model parameters.

underlined by these experiments. All attacks are $\ell_\infty$ within $[0, 1]$ ball. All models were trained on CIFAR10 for 200 epochs, using batch size of 128.

The baseline we could find is as follows. For FGSM White Box atack on CIFAR10, Madry et al. (2018) report 56.1%. For other cases, we could not find baseline figures for these. Therefore, we consider 56.1% as the representative baseline for each of the cases in FGSM.

For PGD, Madry et al. (2018) had reported 50.0% for $k = 7$ steps, and 45.8% for $k = 20$ steps. We performed PGD for $k = 10$ steps on AALR on CIFAR10 on WRN-28-10, and obtained 52.83%. On SimpleNetV1, PGD with 10 steps on CIFAR10 with AALR gives 45.32%.

We use cross entropy loss in all cases (next set of experiments would examine the performance with CW loss for adversarial attacks).

| Model | AALR | SGDR | CLR | ADAM |
|---|---|---|---|---|
| Resnet-18 ($\epsilon = 8/255$, $\alpha = 2/255$) | 66.91 | 59.89 | 68.19 | 33.26 |
| WRN-34-10 ($\epsilon = 4/255$, $\alpha = 2/255$) | 65.86 | 61.55 | 55.63 | 16.13 |
| SimpleNet-V1 ($\epsilon = 8/255$, $\alpha = 2/255$) | 65.02 | 55.4 | 62.43 | 17.88 |
| WRN-28-10 ($\epsilon = 8/255$, $\alpha = 2/255$), 200 epochs | 68.25 | 62.32 | 63.06 | 17.69 |

Table 3: Adversarial Training on CIFAR10: Peak Adversarial Accuracy on FGSM White Box attack, obtained on different models by training with AALR and other adaptive algorithms.

| WRN-28-10(A) | SimpleNetV1(A) | ResNet-18(A) | SimpleNetV1(B) |
|---|---|---|---|
| 52.83 | 45.32 | 50.3 | 31.76 |

Table 4: AALR for adversarial Training on CIFAR10 and 100: Peak Adversarial Accuracy on (A)PGD attack on CIFAR10, (B)FGSM on CIFAR100.

## 5.5 RELATIONSHIP OF BATCHSIZE AND ENTROPY WITH LEARNING RATE

The relationship of learning rate with batchsize for training DNNs is a well-studied topic. In particular, a linear relationship is suggested by Goyal et al. (2017); Smith (2017) and also by Chaudhari & Soatto (2017). The linear scaling rule suggests that the ratio of learning rate to batch size: $\eta/B$ should remain constant for similar generalization across batch sizes for a given dataset and model). This has been exploited to achieve good generalization across batchsizes by Goyal et al. (2017) and Smith (2017). Chaudhari & Soatto (2017) suggest that this ratio should not be too small to maintain the implicit regularization of SGD, otherwise the entropy would go to zero and generalization would be poor. This observation is in line to that by Keskar et al. (2016) where small batchsizes were seen to be essential for good generalization. A square root scaling rule of learning rate and batchsize ($\eta \propto \sqrt{B}$) was suggested by Hoffer et al. (2017).,

We performed a simple experiment (thanks to the suggestion by anonymous reviewers) to observer if AALR can work across batch sizes, adjusting the LR to achieve same generalization. (Note that in the earlier experiments the batch sizes chosen were as per the baseline settings as given in the literature, which already shows the effectiveness of AALR across batch sizes, however this is a more direct experiment to observe the same). We used a different dataset and model than the ones tested earlier: FMNIST on Resnet-20, and tried batch sizes 32, 128 and 512. We obtained similar generalization performance across all the batch sizes. In particular, the peak accuracy were as follows: 95.33 for BS 128, 95.3 for BS 512, and 94.85 for BS 32. (The baseline for the above is 95.63 given by Zhong et al. (2017). On BS 128, we tried this with Random Erasing ($p = 0.5$), and again got similar generalization: 95.47 (reported baseline is 95.98). The corresponding LR trajectory is given in Figure 1. Interestingly, AALR seems to find trajectories that follow the square root rule (with a lot of fluctuations), rather than linear. In particular, if one observes closely around epochs 120 and 200 (places where the LR changes across the batchsizes), this would be apparent. Around 120, BS 32 has LR oscillating between 0.0 and 0.025, BS 128 has LR oscillating between 0.1 and 0.5 and BS 512 has LR oscillating between 0.2 and 0.1. Around epoch 200, for BS 32, LR is between 0.025 and 0.0125, for BS 128 between 0.5 and 0.025, for BS 512, between 0.1 and 0.5.

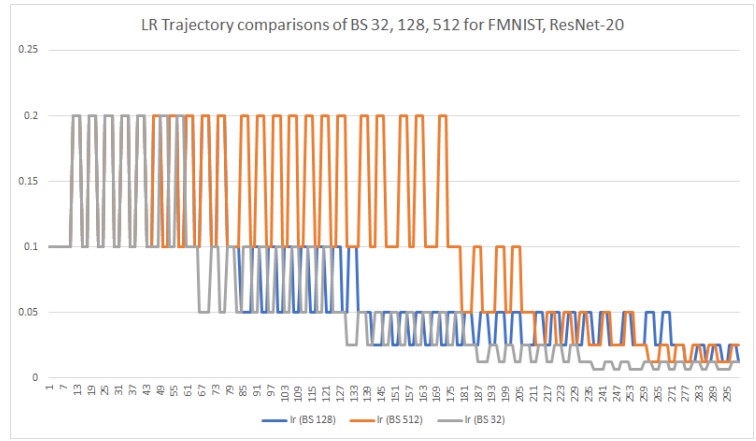

Figure 1: FMNIST, LR trajectory across batch sizes.

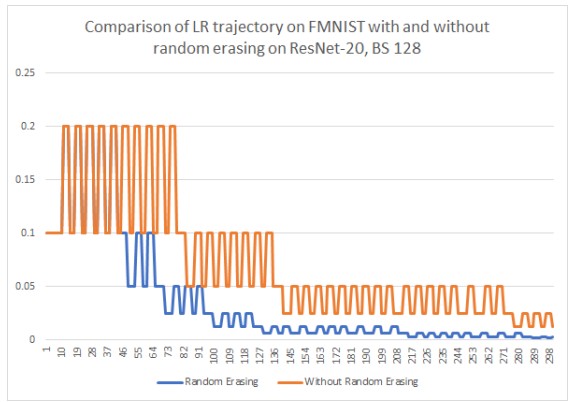

Figure 2: FMNIST, LR trajectory with and without random erasing on same batchsize.

Furthermore, we compared the trajectory obtained on FMNIST, ResNet-20 on the same batch size 128, with and without random erasing (Figure 2). Random erasing is the process of randomly replacing the image pixels with random values for certain rectangular patches with certain probability, commonly used for data augmentation for generating more robust models and improving generalization. Clearly random erasing is a regularizing technique that increases entropy. Interestingly, the LR trajectory shows that AALR is much more conservative in the case of random erasing, in the sense that it maintains a much lower LR compared to the standard case. This fits very well with the observation by Chaudhari & Soatto (2017) that the entropy or implicit regularization in SGD depends on the ratio of $\eta/B$. Since in this case, the entropy or regularization is high due to random erasing, AALR lowers $\eta$ and decreases the implicit entropy of SGD to maintain stability and converge to the similar generalization.

## 6   CONCLUSION

We have presented an autonomous adaptive algorithm AALR that works without tuning across models, datasets, batchsizes and natural or adversarial training to achieve generalization performance comparable or better than SOTA LR-tuned values. Moreover, it is a stable algorithm and consistently converges smoothly, unlike other SOTA adaptive algorithms compared with. This is very promising as it possibly implies that AALR can be reliably applied for new datasets or models or adversarial attacks without requiring extensive manual experimentation. In the future, we would like to try this on tasks other than image classification, and develop a rigorous theoretical justification behind its consistent performance.

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
