# OpenReview forum: "A Simple Dynamic Learning Rate Tuning Algorithm For Automated Training of DNNs"
_ICLR.cc/2020/Conference — Reject_

### Official Review · AnonReviewer2 · 2019-10-20
**Official Blind Review #2**

**Rating:** 1

**Review:**

This paper proposes a new way of scheduling the learning rate in optimization algorithms such as SGD. It is a stand-alone, parameter-free approach that optimistically doubles the learning rate at every loss improvement between two epochs, until the loss increases too much or diverges, in which case the learning rate is divided by two.
This approach is theoretically proven to converge and to follow an optimal scheduling strategy.
In addition, the authors experimentally tested their approach on two image classification tasks, showing that the proposed algorithm yields similar to baseline results.

I am rejecting this paper because it seems to motivate things with non-related facts, experiments are not robust and thorough enough, and there is no conclusion (not even in the appendix).

- The most important thing in this paper to me is the fact that "adversarial training" is used to motivate this approach a lot. it is mentioned 14 times across the paper: 3 times in the abstract alone. Yet there is no explanation of what it is, and how is it different from "natural training" as mentioned in the paper. I suggest the authors either to clearly explain the difference between the two and explain why their approach may help in one setting or the other; or to simply remove the mentions of "adversarial training" if it is not important to the approach.
- to better motivate the approach, I would suggest the authors include different tasks, rather than different training settings. For instance by having one image classification task (keep one of the two current ones) and one text classification or even generation task. This would show that the proposed approach generalizes well to other network architectures.

- The second concern I have is about the experiments. If increasing the learning rate like the proposed approach is making training to convergence faster, then why are the experiments only measuring test set accuracy and not also runtime to convergence?
- Overall, the experiments are not complete and thorough enough: some table values are missing, the set of adversarial training experiments on CIFAR100 are not reported, and some experiments diverged with the ADAM optimizer. Less than 20% accuracy on a 10-class image classification task seems very far from optimal.

- Eventually, I strongly suggest the authors submit a better closing statement than "We use cross entropy loss in all cases." (especially after having read this same sentence earlier in section 5.1 of the paper). No conclusion is added to the paper, not even in the appendix.

Below are a few minor points not taken into account in the scoring but that could make the paper slightly better:
- Section 1, paragraph #1, first sentence: a few citations here would be nice.
- typo on the first line of page 3: "5The pseudocode ..."
- typo in the 2sn paragraph of section 4: "... the convergence is the fast*est* since the step sizes ..."
- typos in the first line of the second paragraph of section 4.1: "Assuming that *the* loss surface is smooth, *the* loss will continue..."
- page 6: "At this time, p=2^{n-1} at this time."
- Section 5.1, paragraph 2, first sentence: "... with dropout and with both dropout and cutout, ..."


**Experience Assessment:**

I do not know much about this area.

**Review Assessment: Checking Correctness Of Derivations And Theory:**

I assessed the sensibility of the derivations and theory.

**Review Assessment: Checking Correctness Of Experiments:**

I carefully checked the experiments.

**Review Assessment: Thoroughness In Paper Reading:**

I read the paper thoroughly.

---

> ### Author Response · Authors · 2019-11-14
> **Addressing the major comments**
>
>
> Thank you for the insightful review and comments.
>
> 1. We agree that we have tried AALR on image classification, both natural and adversarial training, and we should try it on other tasks too. This is definitely our plan for future work.
>
> 2. Yes we could measure the time to convergence. AALR converges around the same time as the LR tuned baselines. However, in case of the other periodic methods like SGDR and CLR, measuring the convergence becomes very difficult (in fact detecting the convergence is difficult) due to their oscillating nature, hence we have not reported these values.
>
> 3. We have included a Conclusion.
>
> 4. We have also added details on adversarial versus natural training principles on section 5.4.
>
> 5. We have added the missing values. We have also started experiments on adversarial training of CIFAR100 and reported values for those that are complete by this revision. Additionally, as pointed out by the other reviewers, we have started additional experiments on Hypergrad etc. We have reported the values for runs that are complete by the time of this revision.
>
> 6. The catastrophic failure of ADAM on multiple cases is a drawback of the method, unlike the proposed algorithm AALR that works uniformly well. In fact, we performed an experiment with AALR on top of ADAM optimizer for one of the cases where ADAM failed. It considerably improved the performance of ADAM from 10% to 78.46%, again showing the effectiveness of AALR.
>
> 7. We have corrected the typos, and have added the citations in the Introduction.

---

### Official Review · AnonReviewer3 · 2019-10-22
**Official Blind Review #3**

**Rating:** 1

**Review:**

The paper considers the problem of automated adaptation of learning rate during (deep) neural network training. The use cases described are standard and adversarial training for image classification. Given the wide use of DNNs in computer vision (and other areas), learning rate tuning is clearly an important problem and is being actively researched.

The proposed learning rate adaptation procedure consists of a straightforward combination of learning rate halving/doubling and model checkpointing. Experimental results from implementing the adaptive learning rule for standard and adversarial training on CIFAR are provided. Multiple architectures are tested in each setting. The paper claims a primary advantage of the proposed learning rule to be that it requires no tuning as opposed to other rules such as SGD, Adam.

My decision is to reject the paper due to methodological issues with the experiments and lack of evidence wrt/ dataset variety. The paper should be considered a work-in-progress that may have potential in a more focused setting, e.g., adversarial training as described in the paper.

***

The major claim of the proposed algorithm not requiring any manual tuning is technically true but misleading. The algorithm does have parameters (SGD momentum, batch size, initial learning rate, patience) with values that were set somehow. In fact, a major methodological issue with the experiments is that the reader does not know if the datasets were used to both set these values and to assess performance, i.e., there are no obvious "held-out" datasets. Also, there is no rigorous or even informal justification of the settings. It could be that the paper is arguing that the specific values will result in competitive, if not better, performance than baselines across a variety of datasets - unfortunately, only two datasets are utilized in the experiments, and one, CIFAR10, is not considered challenging. This leads to the second issue with the paper: the experimental validation is not extensive wrt/ datasets which is significant given that the form of the evidence for the proposed method is almost entirely empirical.

Additionally, I don't agree that competitor algorithms should not be tuned b/c the proposed method does not require tuning. Even if the proposed method does not require tuning (as stated previously, I don't believe this to be accurate), that does not imply a fair comparison precludes tuning competitors via, e.g., cross validation. The only relevant quantities are final test-set performance and total training time/resources required.

The well-known interdependence between learning rate and batchsize as noted in e.g., Hoffer et al. (2018), is not addressed by the experiments. Batchsizes in the experiments vary, but no justification is provided for how these are selected.

Finally, the paper is unfinished as some experimental runs were not complete at the time of submission.

On the positive side, the general point about the necessity of learning rate tuning for adversarial training (described in the fourth paragraph of the introduction) is a very good one, and there may be an opportunity for a more focused application of the proposed algorithm perhaps among further datasets and considering additional, alternative attacks.

***

Suggestions for improvement / questions (related to decision):

* It should not be a challenge to find more image classification datasets to include in the experimental comparison: SVHN, Fashion MNIST, Imagenet, ... Using these, the paper can either follow the standard train/test methodology *across datasets*, i.e., split the meta-dataset into train/test, and/or provide a more compelling body of evidence for the proposed method. Also, the performance dependence on batchsize amongst the proposed algorithm and competitors should be investigated experimentally.

* The Baydin et al. (2018) algorithm should be added to the set of competitors since it would provide a relatively easy* reference point wrt/ "hypergradient" approaches. I don't agree with the statement in the related work section that this entails "additional computation of gradients." *In the sense that the rule should be straightforward to implement.

* The convergence analysis assumption that the optimal oracle SGD follows typical learning rate regimes motivated by loss plateauing seems to be in direct contradiction to the sentiment expressed in the cited Hoffer et al. (2018) paper that such "rules of thumb" may be misguided. Can the authors discuss the appropriateness of their assumption wrt/ this point? Also, in the convergence analysis, the phrase "in expectation" is used twice. This has a specific probabilistic meaning, but appears to be used heuristically in this section. Can the authors clarify whether this usage is informal or formal? If the latter is true, it would be better to provide a more formal convergence argument that explicitly takes the inherent randomness into account.

***

Editorial comments (not related to decision):

* Introduction: The first two sentences of the second paragraph, particularly the second, would do well to have an accompanying reference or references.

* Proposed method: Even as an informal statement, the second sentence of the second paragraph under the Phase 2 sub-heading is problematic. The proposed method does not "resist" lowering the learning rate "for as long as possible" so much as it doesn't lower the learning rate for a fixed number of epochs (algorithm parameter).

* "Adversarial training" section (5.4): The paper assumes the reader is familiar with the terms "FGSM", "white box", and parameters \epsilon and \alpha since these are referred to w/o description. Perhaps a short (2-3 sentence) description of the adversarial scenario could be added?

* Experiments: It would be good for the paper to include RMSProp and Adagrad results to the experimental tables as these rules are both readily available for use and widely used.

* Experiments: Is the reporting of the peak accuracy standard in the literature?

* Experiments: I want to give the paper credit for performance on CIFAR100, but this is difficult without explicit points of comparison. This can be easily remedied by including SOTA performance values (along with appropriate references) in the tables or text.

* (Potential) Typos:
	Proposed method algorithm description:
		Requirements has a weight decay parameter which seems strange given that the algorithm is performing automated learning rate adaptation...
		The epoch counter is incremented in line 5, but not reset prior to Phase 2. Does this mean that Phase 1 training epochs are counted toward the total (T)?
		Line 7 should be \eta_t <- \eta_0 / 2.
		The patience counter in line 15 is not utilized below.
		Line 23 could/should be an else statement.

**Experience Assessment:**

I have read many papers in this area.

**Review Assessment: Checking Correctness Of Derivations And Theory:**

N/A

**Review Assessment: Checking Correctness Of Experiments:**

I assessed the sensibility of the experiments.

**Review Assessment: Thoroughness In Paper Reading:**

I read the paper thoroughly.

---

> ### Author Response · Authors · 2019-11-14
> **Addressing the major comments**
>
>
> Thank you for the insightful and thorough comments and review.
>
> 1. We have tried on a different dataset FMNIST as suggested.
>
> 2.We have experimentally verified the performance dependence of AALR on batchsize. We found that the generalization performance of AALR remains unaffected wrt batchsize. A more detailed discussion on this has been provided as an official comment above. The new results and observations are presented in the revised version Section 5.5
>
> 3. We have implemented SGDHD algorithm of Hypergrad (Baydin et al.). We have reported the results of runs that are already finished. Other runs are on the way. We will add these values as soon as they are complete.
>
> 4. Hoffer et al. had pointed out that “rules of thumb” like linear scaling of learning rate with batchsize may not hold in every case. We agree that there needs to be a solid theoretical founding behind every such rule. However, for ease of analysis we have assumed that OPT behaves like a typical LR regime on image datasets, that typically obeys the stated rule of thumb.
>
> 5. We agree the usage of "expectation" was not rigorous or formal and have removed the usage from the text.
>
> 6. The references for the baseline values, including those obtained on CIFAR100 are reported in section 5.3.
>
> 7. We have added some details explaining the concepts of adversarial training like FGSM and PGD in section 5.4.
>
> 8. We have changed the description in Phase 2 to take into account the reviewer’s comments.
>
> 9. Weight decay: it is a regularization parameter that can be optionally provided by the user. This does not refer to the factor by which LR is adjusted by AALR.
>
> 10. Epoch counter: yes the epochs in Phase 1 are *counted* in the total number of epochs (T).
>
> 11. We will include the comparison with Adagrad and RMSProp in the camera ready version, if accepted. However, in general, it has been observed in practice that ADAM achieves a slightly better generalization compared to these.
>
> 12. Tuning of competitor algorithms: We agree that the relevant metrics are test set performance and training time and resources required. All the algorithms tried for any given dataset-model-batchsize were run for the same number of epochs, thereby approximately same training time and resources were allocated to each. Hence, this did not leave any time/resources for the manual tuning part.
>
> In any case, our aim with AALR is to present an autonomous approach that works without tuning across tasks, eliminating the need for manual, exhaustive experimentation.

---

### Official Review · AnonReviewer1 · 2019-10-28
**Official Blind Review #1**

**Rating:** 1

**Review:**

This paper proposes an algorithm for automatically tuning the learning rate of SGD while training deep neural networks. The proposed learning rate tuning algorithm is a finite state machine and consists of two phases: the first phase finds the largest learning rate that the network can begin training with for p = 10 epochs; the second phase is an optimistic binary exploration phase which increases or decreases the learning rate depending upon whether the loss is NaN, increasing or decreasing. Empirical results are shown on a few standard neural networks for image classification on CIFAR-10/100 datasets and for adversarial training on the CIFAR-10 dataset.

I recommend rejecting this paper for the following reasons: (i) the algorithm developed here is extremely heuristic, no insight, theoretical or empirical, is provided as to why this could be a general algorithm, (ii) a major claim in the paper is that the automatic learning rate tuning does not have any hyper-parameters but the actual algorithm does have parameters such as patience and successive doubling of the learning rate although they are tuned adaptively using ad-hoc heuristics, (iii) the convergence analysis is not at all rigorous, in particular the optimal oracle for SGD  may not exist, and (iv) the baseline algorithms are not tuned and the minor improvements of the proposed algorithm over them is therefore not significant.

Some questions that I would like the authors to answer:

1. While the first phase of the algorithm seems a reasonable thing to do, the second phase is full of heuristics which I am not sure will work well for all problems. For instance, I do not see why the algorithm performs trains for p epochs twice even if the loss increased after the first stage, or why the learning rate should be increased if the loss decreased after the second stage.
2. Section 4, bullet 3/4 in the definition are problematic: the loss in SGD is not monotonically decreasing with respect to time. What does divergence of training mean here? What does “Any SGD algorithm” mean? Do you instead mean any first-order stochastic gradient-based algorithm?
3. If you imagine a double well potential with one wide minimum and one sharp minimum, both at the same training loss, if OPT starts in the sharp valley, it will not be able to go to the wide valley without the training loss increasing.
4. Have you tried this algorithm on other problems which are sensitive to the values of learning rate, e.g., training optical flow or segmentation networks?
5.  The wordy and heuristic argument in Section 4.1 rests on statements like “AALR and OPT arrive at roughly the same location after so and so epochs and hence reaches similar generalization performance”. This cannot happen in a non-convex landscape, the trajectory of SGD starting from the same initial condition can be very different across two independent runs. Therefore, I also don’t see why the latter half of the statement about generalization should be true.
6. Can you make the development in Section 4 rigorous?
7. Why are some runs for SGDR stuck at 10% accuracy in Table 1-2?
8.  FGSM is a very weak attack for measuring adversarial accuracy. Can you show results with a better attack, say a few steps of PGD?


Some suggestions to improve the paper:

1. A simple experiment to check the automatic tuning would be to increase the batch-size of the same network while maintaining the ratio of batch-size and learning constant (see https://arxiv.org/pdf/1706.02677.pdf, https://arxiv.org/abs/1710.11029, among others). It would be interesting to see whether the auto-tuner finds a learning rate that corresponds to stable learning without degradation in the generalization performance.

**Experience Assessment:**

I have published in this field for several years.

**Review Assessment: Checking Correctness Of Derivations And Theory:**

I carefully checked the derivations and theory.

**Review Assessment: Checking Correctness Of Experiments:**

I carefully checked the experiments.

**Review Assessment: Thoroughness In Paper Reading:**

I read the paper thoroughly.

---

> ### Author Response · Authors · 2019-11-14
> **Reason for persisting in a higher learning rate; increasing LR when loss decreases.**
>
> The reason for persisting in the learning rate even when the loss increases in the first stage is to handle scenarios such as the double well potential as the reviewer has pointed out in question 3. In order to escape a sharp minima, the loss might first increase for a few rounds, before starting to decrease again. Hence, reverting to a lower learning rate at the first instance of loss increase might not help in escaping from a sharp minima. To handle such situations, we have designed AALR to persist in the higher learning rate for p more epochs.
> We increase the learning rate when we see a decrease in training loss for two reasons: a) it might be a sharp minima and we want AALR to explore the vicinity of the loss valley (in particular, escape if it is a sharp minima); b) if we are in a smooth, flat loss well, increasing the LR will likely not harm, and it will simply accelerate AALR to reach the minimum point, without escaping from the loss valley.

---

> ### Author Response · Authors · 2019-11-14
> **Inconsistency in definition; double well potential**
>
> Thank you for pointing out the inconsistency in the existing definition.
> Divergence of training means here: if the LR is set to any larger value than that of OPT, gradients will explode and the training of the network cannot be recovered from there. Yes “Any SGD algorithm” refers to any first order stochastic gradient-based algorithm. We have edited the text to reflect these corrections.
>
> Double well potential: We addressed this point earlier in the comment on the reason for persisting in higher LR and increasing LR when loss decreases. OPT would know the exact LR which would help it to escape from the sharp minima without leading to divergence due to exploding gradients. AALR is designed to handle such situations. For ease of analysis, we have assumed we have a reasonably smooth loss surface where a typical SGD regime (where LR never increases) is a proxy for OPT and the loss will show a decrease after p (\geq 2) epochs on being trained at the tuned LR.

---

> ### Author Response · Authors · 2019-11-14
> **Low values of SGDR; PGD instead of FGSM; Informal convergence; other tasks**
>
>
> Low values of SGDR: We don’t know, it is probably due to some inherent behaviour of the algorithm SGDR that leads to its catastrophic failure in certain cases. In fact, we would like to add that even ADAM failed catastrophically in a few cases. We have performed an experiment where we applied AALR on top of ADAM in one such case. The accuracy improved from 10% to 78.46%.
>
> PGD vs FGSM: We have added some results on using AALR for adversarial training with PGD attack (10 steps).
>
> Informal convergence: We agree the development in section 4 is not completely rigorous. We have removed the statement on generalization.
>
> Other tasks: We haven't tried AALR on the proposed tasks yet. That is definitely the next step in future work.

---

> ### Author Response · Authors · 2019-11-14
> **batchsize vs LR**
>
> Thank you for the insightful review and detailed comments and suggestions.
>
> We have performed experiments to verify this relationship. The detailed comment can be found as an official comment above this. The new results can be found in the revised version of the paper in the last Section before Conclusion.
>
> We have tried to address all other comments below.

---

### Author Response · Authors · 2019-11-14
**Relationship between LR, batchsize and entropy.**

We thank the reviewers for their careful review and kind, insightful comments and pointers.

The reviewers wanted to know if AALR can find the relationship between LR and batchsize and they wanted us to try on datasets other than CIFAR10 and CIFAR100.

We performed additional experiments to examine how AALR handles different batchsizes and if it can find the relationship between learning rate, batchsize and entropy. (We would like to highlight here that for all the other experiments we used the settings including batchsizes as used in the baselines reported in the literature. The baseline sources are all provided in the section 5.3).

We tried on FMNIST (since it was recommended to try on datasets other than CIFAR10 and CIFAR100) on different batchsizes. We found that AALR finds LR trajectories that roughly obey the *square root* relationship between LR and batchsize (and not linear).

Moreover, we compared the LR trajectory as obtained on the same batchsize with and without random erasing. We found that the LR trajectory is much more conversative (i.e. low LR) for the one with random erasing. This is in line with the observation in  https://arxiv.org/abs/1710.11029 (pointed out by Reviewer#1) that the implicit entropy in SGD is determined by the ratio of LR to Batchsize. Since with random erasing regularization the entropy increases, the implicit SGD entropy is lowered by AALR by reducing the LR.

These new experiments and observations have been presented in the revised version in Section 5.5. We would like to mention that in all of these cases, AALR’s generalization is unaffected, it performs similar to reported baseline values, across batchsizes.

---

### Decision · Program_Chairs · 2019-12-19

**Decision:**

Reject

**Comment:**

This paper proposes an automatic tuning procedure for the learning rate of SGD. Reviewers were in agreement over several of the shortcomings of the paper, in particular its heuristic nature. They also took the time to provide several ways of improving the work which I suggest the authors follow should they decide to resubmit it to a later conference.